# Mercury Pollution from Artisanal and Small-Scale Gold Mining in Myanmar and Other Southeast Asian Countries

**DOI:** 10.3390/ijerph19106290

**Published:** 2022-05-22

**Authors:** Pyae Sone Soe, Win Thiri Kyaw, Koji Arizono, Yasuhiro Ishibashi, Tetsuro Agusa

**Affiliations:** 1Graduate School of Environmental and Symbiotic Sciences, Prefectural University of Kumamoto, Kumamoto 862-8502, Japan; 2Faculty of Environmental and Symbiotic Sciences, Prefectural University of Kumamoto, Kumamoto 862-8502, Japan; yisibasi@pu-kumamoto.ac.jp (Y.I.); te-agusa@pu-kumamoto.ac.jp (T.A.); 3Research Institute for Humanity and Nature, Kyoto 603-8047, Japan; thiri@chikyu.ac.jp; 4Graduate School of Pharmaceutical Sciences, Kumamoto University, Kumamoto 862 0973, Japan; arizono@kumamoto-u.ac.jp

**Keywords:** Hg, artisanal and small-scale gold mining, air, water, soil, plant, fish, human hair, health risk, Myanmar, Southeast Asia

## Abstract

Mercury (Hg) is one of the most harmful metals and has been a public health concern according to the World Health Organization (WHO). Artisanal and small-scale gold mining (ASGM) is the world’s fastest-growing source of Hg and can release Hg into the atmosphere, hydrosphere, and geosphere. Hg has been widely used in ASGM industries throughout Southeast Asia countries, including Cambodia, Indonesia, Laos, Malaysia, Myanmar, the Philippines, and Thailand. Here, 16 relevant studies were systematically searched by performing the PRISMA flow, combining the keywords of “Hg”, “ASGM”, and relevant study areas. Mercury concentrations exceeding the WHO and United States Environmental Protection Agency guideline values were reported in environmental (i.e., air, water, and soil) and biomonitoring samples (i.e., plants, fish, and human hair). ASGM-related health risks to miners and nonminers, specifically in Indonesia, the Philippines, and Myanmar, were also assessed. The findings indicated severe Hg contamination around the ASGM process, specifically the gold-amalgamation stage, was significantly high. To one point, Hg atmospheric concentrations from all observed studies was shown to be extremely high in the vicinity of gold operating areas. Attentions should be given regarding the public health concern, specifically for the vulnerable groups such as adults, pregnant women, and children who live near the ASGM activity. This review summarizes the effects of Hg in Myanmar and other Southeast Asian countries. In the future, more research and assessment will be required to investigate the current and evolving situation in ASGM communities.

## 1. Introduction

### 1.1. Mercury

Mercury (Hg) is listed among the top 10 most harmful metals by the World Health Organization (WHO), and its chemical forms are considered a public health concern [1]. All its common forms, including elemental (metallic), inorganic, and organic are highly toxic. In particular, methylmercury (MeHg) is the most dangerous form because it can bioaccumulate in microorganisms and biomagnify or enhance the trophic levels in aquatic food webs [2]. Meanwhile, elemental Hg can be converted to MeHg in aquatic sediments. The use of elemental Hg in artisanal and small-scale gold-mining (ASGM) sector can be hazardous because of the inhalation of Hg vapor, which easily penetrates the blood–brain barrier and induces neurotoxicity [3]. A famous catastrophic Hg outbreak occurred in Minamata Bay, Japan, in the 1950s, when factory wastewater containing MeHg from a factory was discharged into the Shiranui Sea, poisoning the people who ingested the contaminated seafood [3]. This became one of the first and the most critical incidents of Hg poisoning due to an industrial site.

Different forms of Hg can be released into the atmosphere, water, and across land as a result of human activities such as burning of fossil fuels (e.g., coal and petroleum), industrial effluents, product waste (e.g., electronic) from intentional use, dental amalgamation, agricultural practices, and ASGM and natural processes, including volcanic eruptions, rock weathering, and forest fires. Thus, Hg is discharged worldwide into the environments. ASGM is the world’s fastest-growing source of Hg and can discharge Hg into both the aquatic environment and the terrestrial ecosystem. The emission of Hg into the atmosphere via ASGM account for 37.7% of global outputs among the other Hg emission sources, with South America, Asia, and Sub-Saharan Africa as the primary sources [4]. Meanwhile, ASGM occurs in more than 70 nations worldwide, with an estimated 16 million miners who afford up to 20% of the annual global gold production [4]. According to the global inventory of the United Nations Environmental Program (UNEP) in 2015 [4], 2220 tons of Hg was emitted to the atmosphere from all anthropogenic sources. Notably, 38% (838 tons) of Hg was emitted from the ASGM sources (Figure 1).

### 1.2. Hg Use in ASGM Communities

The use of Hg is quite common in ASGM sectors worldwide, including Southeast Asian countries such as Cambodia, Indonesia, Laos, Myanmar, the Philippines, and Thailand. The earliest records of Hg use in alchemy and amalgamation were from Egypt and China more than 3000 years ago [5]. Hg has been used in inexpensive, easy, and rapid approaches for extracting gold from its ore and soil [6]. Notably, owing to weak legislation, poor engagement, contribution of artisanal miners, and easily accessible of robust black market for Hg usage in the ASGM will continue to persist [7,8,9].

Myanmar, a developing country in Southeast Asia, has various mineral and natural resources, such as for jade, gold, ruby, and copper. Myanmar has been facing overexploitation of natural resources for more than two decades because its people seek to extract its natural resources illegally [10]. Additionally, 70% of the population of Myanmar is in rural areas and rely on natural resources. Meanwhile, gold production in Myanmar has increased slightly in recent years, reaching 1700 kg in 2016 [11]. ASGM activities in this country have been conducted throughout the states and regions of the country, namely Bago, Kachin, Mandalay, Mon, Sagaing, Shan, and Tanintaryi (Figure 2) [12]. 

Various ASGM processes can be categorized on the basis of region and depend on the gold deposit types. ASGM can be classified using several methods such as panning, river mining with bucket dredges, suction dredging, and hydraulic mining. Alluvial deposits (river sediments) and hard-rock deposits (typically gold in quartz veins) have been exploited the gold ore by local and migrant miners. In the most common method for ASGM in Myanmar, gold ore is excavated via underground or open-pit mining, after which it is dried and ground using a powdering machine. The resulting powder is placed in a pan containing water to separate gold particles via gravity settling. Gold particles are collected at the bottom of the pan together with sand. A piece of Hg (like a finger-tip) is added to the pan to extract gold by forming a gold–Hg amalgam, which is squeezed by hand through a fabric cloth. Subsequently, a mine operator vaporizes the Hg in the gold-amalgam using a burner to obtain pure gold. Hg vapors are consequently released into the atmosphere and deposited into aquatic and terrestrial ecosystems. The ASGM practices in the considered study areas are summarized in Figure 3 and Figure 4.

### 1.3. Minamata Convention on Hg

The Minamata Convention on Hg became one of the first worldwide environmental agreements in the 21st century. The convention was adopted in 2013, and to date, 123 countries have signed the agreement. The convention aims “to protect human health and the environment from anthropogenic emissions and releases of Hg and the compounds and it sets out a range of measures to meet the objective” [13]. According to Article 7 of the Minamata Convention, each party from an ASGM shall develop a national action plan regarding Annex C, which indicates implementing national objectives and reducing targets and actions to eliminate the Hg and related compounds. In the national action plan (NAP) for ASGM sectors, the informal ASGM sector must be regulated to accomplish the requirements for reducing the Hg in the country. Key to an NAP is the development of Hg inventories and baselines in the ASGM sector to monitor improvements and establish regulatory standards for Hg emission reduction. The pertinent parties must cooperate with the relevant stakeholders from governments, industry, NGOs, and academia. Subsequently, the parties need to build awareness regarding all Hg compounds in an ASGM process, promote non-Hg alternative practices, and provide technical and financial support. The countries that have not ratified the Minamata Convention in the Association of Southeast Asian Nations (ASEAN) include Brunei, Laos, and Myanmar. However, in Myanmar, the NAP for Minamata Initial Assessment, which is funded by a global environmental facility, has started a national Hg inventory. The status of the Minamata Convention in ASEAN countries is summarized in Table 1.

### 1.4. Objective

Mercury pollution is a worldwide problem especially in ASGM countries. The number of artisanal miners has increased over the years and now totals to ~45 million people [15], with at least half of them engaged in gold mining, extracting up to 450 tons of gold per year in at least 70 countries [16]. ASGM activities produce increasing amounts of gold from countries in Africa (e.g., Ghana, Mali, Sudan, Tanzania, and Zimbabwe), Latin America (e.g., Brazil, Colombia, and Peru), and Asia (e.g., China, Indonesia, Mongolia, Myanmar, and the Philippines).

This study emphasized Myanmar and other ASEAN countries that are practicing ASGM. Similar to other developing countries, environmental challenges in Myanmar have been given strong consideration since natural resources have been extracted by illegal measures. Achieving an environmental balance has become a crucial role for such challenges. Unfortunately, Myanmar has insufficient professional labors with acquired skills; an ineffective governing mechanism; no transparency for trading by-products of gold, specifically Hg; and minimal research activities. Therefore, there are only six publications on Hg pollution in Myanmar up to now [17,18,19,20,21,22].

This review has identified and assessed the critical Hg pollution issues in Myanmar and other Southeast Asian countries. This study outlines the Hg problems that are of crucial concern to the citizens of these nations. We evaluated the Hg contamination in environmental media and the risks of Hg exposure on human health and propose a relevant policy framework regarding Hg issues.

## 2. Materials and Methods

### 2.1. Study Selection

The study identified the relevant literature published between 2000 and 2021 using databases including *PubMed*, *Web of Science*, *Springer*, *Science Direct*, and *Google Scholar.* The keywords used during the search were “Hg”, “ASGM”, “Myanmar”, “Indonesia”, “the Philippines”, and “Malaysia”. Research materials on international regulations, laws, and procedures related to Hg problems were also considered. The study focused on research articles from Myanmar and other ASEAN countries. To conduct the screening study, two reviewers (P.S.S. and T.A.) compared the titles and abstracts of the studies according to the inclusion and exclusion criteria presented in Table 2.

### 2.2. Quality Assessment 

In selecting the literature, the study focused on the Preferred Reporting Items for Systematic Reviews and Meta-Analyses (PRISMA) statements [23] on identification, and screening, and included research as shown in Figure 5. The search method considered mainly published literature on Hg-related studies in environmental science, social science, and public health. The study focused on original research articles and systematic reviews. To ensure the quality of the evaluation, duplications were evaluated and checked rigorously. In the exclusion criteria, we considered the publication year (2000–2021) the language used in the research articles. Only research published in English from the study areas in Southeast Asian countries were included. Third-party tools (e.g., Microsoft Excel and Mendeley) were used for importing website data and data screening.

## 3. Results

### 3.1. Hg Concentrations in Air

Hg in the atmosphere occurs primarily in three forms, namely the gaseous state of elemental Hg (Hg(0)), reactive gaseous Hg (Hg(II)), and total particulate Hg (Hg(p)) [24]. Hg(0) emission from ASGM activities is the highest Hg emission source. Hg vapor, (mainly in the chemical form of elemental Hg(0)) can travel vast distances in the air and be deposited or captured in forest treetops and leaves [25]. According to the 2018 global Hg assessment report, the global emission of Hg into the air in 2015 from ASGM sources was 838 tons, with East and Southeast Asian countries accounting for ~214 tons [4]. Recent studies have showed very high Hg concentrations in the atmosphere resulting from ASGM activities in Central Sulawesi, Indonesia [26], Camarines Norte, the Philippines [27], and Mandalay region, Myanmar [21]. The Hg concentrations in air from ASGM activities in Indonesia, Myanmar, and the Philippines are summarized in Table 3.

In Central Sulawesi, Indonesia, the highest average concentrations of 24 h ambient Hg(0) of 9172 ng/m^3^ were found in the gold-processing areas that refined gold (including the stages of Hg amalgam burning) [26]. This total value was nine times higher than the WHO guideline limit of 1000 ng/m^3^ [28]. Further, this study also considered indoor and outdoor air Hg(0) concentration in the Palu city area and the village of Mangkahui. The highest indoor and outdoor air concentrations of Hg(0) in the Palu city were 450 and 2250 ng/m^3^, respectively. In the village of Mangkahui, the Hg(0) concentrations in the indoor and outdoor air were 196 and 103 ng/m^3^, respectively, at site A. Meanwhile, the values were 238 and 279 ng/m^3^, respectively, at site B.

A study investigated the atmospheric Hg concentration at an ASGM site in the Mandalay region of central Myanmar via two surveys [21]. In the first and second survey, the highest Hg concentrations of 10,900 and 74,000 ng/m^3^, respectively, were noted in an amalgamation-burning area of an ASGM site. These values were several times higher than the Hg limit value in the WHO guidelines [28]. In addition, the study suggested that Hg was dispersed not only in the ASGM areas but also in nearby residential areas.

Atmospheric Hg pollution has been identified in the Province of Camarines Norte, the Philippines, by Murao et al. [27]. The authors focused on an area of a rod-mill station in the ASGM that recently burned gold amalgamation. The highest Hg concentration in air was 314,000 ng/m^3^, which was considerably higher than WHO guideline (1000 ng/m^3^) [28] at the rod-mill station in Benit, the Philippines. Meanwhile, the lowest concentration was 7.8 ng/m^3^ at the same place 4 weeks after the burning. 

In comparison with other air constituents, gaseous Hg(0) is relatively inert. Hg(0) from both anthropogenic and natural emissions can be transported over large distances by air and stay in the atmosphere for a year; therefore, Hg(0) can deposit in terrestrial and aquatic ecosystems [29,30]. The studies from Indonesia, Myanmar, and the Philippines have revealed that the atmospheric concentration of Hg were much higher in the ASGM areas that burn gold amalgamation. The concentrations were also higher than the WHO guidelines [28]. Therefore, a Hg recovery method should be considered in the ASGM industries. 

### 3.2. Hg Concentrations in Water Bodies

Water resources can be impacted by various ASGM operation steps, such as mined-ore sifting and washing. An amalgamation process used in ASGM sectors typically discharges wastewater into water bodies. Subsequently, aquatic organisms are exposed to elevated Hg levels. Furthermore, inorganic Hg can be transformed into toxic MeHg [31]. MeHg in aquatic organisms is biomagnified through the food chain. Notably, fish intake is the primary source of Hg exposure in humans [32]. Mercury concentrations in waterbodies of Indonesia, Malaysia, Myanmar, the Philippines, and Thailand are presented in Figure 6.

A study in the Cikaniki River, Bogor, Indonesia, reported Hg concentrations in the river water ranging from 0.4 to 9.4 µg/L. The highest concentrations were found near an ASGM village. In this study, significant correlations were observed between Hg(0) and MeHg since MeHg concentration was considerably lower than Hg(0). The fact assumes that mining wastes was not a direct source of MeHg in the Cikaniki River [33]. Otherwise, Hg(0) deposited in river water can be subjected to methylation, suggesting that the change in chemical forms of Hg in water systems should be conducted in the future. An earlier study of the river conducted in 2009 reported Hg concentrations of 0.09–9.1 µg/L [34] and showed similar results, indicating the continuous pollution of the Cikanaki River.

In a study conducted in the Mandalay region, Myanmar, two groundwater samples were collected from five ASGM areas [21]. Irrawaddy River is one of the main rivers in Myanmar, which is located near an ASGM area. Surface water samples from upstream and downstream areas of this river were collected. Hg concentrations were also determined in groundwater at the nearby residential areas. The Hg concentrations in the sampled groundwater were in the range of 0–0.04 µg/L. The area closest to gold-mining activities typically showed Hg contamination [35]. The samples obtained in the Irrawaddy River, which was downstream from the ASGM area, showed a Hg concentration of 0.005 µg/L Hg, and samples taken from the upper stream of the river contained 0.004 µg/L of Hg. The reported Hg concentrations in the Irrawaddy river were slightly higher than the typical concentrations of Hg in lakes and rivers (0.001–0.003) [36] but not exceptional.

Interestingly, the Hijo River in the Philippines supports food security and is a means of living for local people who participate in it for gold processing. Wastewater containing Hg and cyanide from mine tailings is discharged into the river without treatment [36]. Additionally, the Naboc River in the Philippines receives effluent water from mining operations. The Hg concentrations found in the Hijo, Naboc, and Kingking Rivers were 78.4, 72.8, and 75.2 µg/L, respectively [37]. These levels are much higher than the national standard limit (i.e., 1 µg/L) of the Philippines [38,39].

A study in Phichit Province, Thailand, focused on the surface water of the Klong Dai Nam Khun and Klong Sa Luang canals, which were connected with a mining area [31]. Thirteen locations with aquatic habitats, including upstream, downstream, reservoir, and other water bodies near a gold-processing area and separatory ditch that were used for gold-ore separation processes, were considered. The study emphasized evaluation of the Hg concentrations level-related Hg-contaminated sites and their distances from ASGM sites, showing Hg concentrations in the range of 0.6–5.4 µg/L. The workplace area showed a higher Hg concentration than areas at greater distances upstream and downstream because the amalgam processes were conducted near the sampling locations.

In the considered studies, the studied samples from Indonesia [33], the Philippines [37], and Thailand [31] exceeded WHO guideline limit of 0.5 µg/L of Hg [40]. ASGM areas could affect Hg concentrations in surface water because Hg can be derived from atmospheric sources [41]. Further, Hg can transfer to the food chain in aquatic environments. Therefore, the effluent from improperly treated wastewater can be detrimental to marine life and people who consume seafood.

### 3.3. Hg Concentrations in Soil

Soil is a key indicator for monitoring environmental Hg concentration because Hg entering the atmosphere from amalgamation burning can deposit in the top layers of soil. It is necessary to understand the level of Hg concentration to prevent Hg pollution of soil. For the topsoil profile, Hg concentrations have been found to decline from the top soil to the deeper horizon [42,43]. In addition, Hg sorption from the air may contribute to Hg accumulation in topsoil horizons through litter accumulation and decomposition. The sources of Hg contamination of soil are fertilizers, lime, sludges, and manures [44]. A summary of the Hg levels in soils from ASGM sites in Indonesia, Myanmar, the Philippines, and Thailand is shown in Figure 7.

An Indonesian study classified forest and paddy field soils impacted by ASGM activities [33]. The Hg concentrations analyzed in forest soil and paddy field soils were 0.07–16.7 and 0.4–24.9 μg/g, respectively. Additionally, the concentrations of MeHg were in the range of 0.07–2 μg/kg in forest soils and 0.07-56.3 μg/kg in paddy field soils. These data demonstrated that paddy field soil is particularly affected by ASGM activities [33].

ASGM activities are widely performed in the upper part of Banmauk Township, Sagaing Region, Myanmar. A study investigated soil samples from the placer gold-deposition area and identified soil matrices based on ASGM operation processes, such as ore processing, sluicing, panning, and amalgamation [17]. The soil matrix from the amalgamation process exhibited the highest Hg concentration of 77.44 μg/g, whereas Hg concentrations during soil ore processing, sluicing, and panning stages (gold-amalgamation stage) were 0.68, 0.51, and 4.86 μg/g, respectively [17].

A study conducted in the Philippines determined Hg concentrations in soils obtained from potentially contaminated hotspots and the areas distant from such spots. The highest Hg concentrations observed were 71.75 μg/g in the sample from a rod-mill station in the amalgamation-burning workplace. By contrast, the lowest concentration observed was 0.15 μg/g in the sample from a nonmining area, showing that the higher Hg concentrations had contaminated the vicinity of the ASGM area [27].

A study of an ASGM operation in the Phichit region, Thailand, considered surface soil (0–5 cm depth) from mining and remote areas. The Hg concentrations in the mining and remote areas were in the range of 0.14–10.56 and 0.038–0.632 μg/g, respectively. The higher Hg concentration observed in the mining area indicated that Hg vapor emitted into the atmosphere was likely deposited on soil surfaces near the burning stoves. This was because of the 7.8 h/day amalgamation process takes place for extraction of 60–150 g of gold [31].

A study reported that Hg concentrations in soil do not typically exceed 0.1 μg/g, and normal levels in soil were reported. Moreover, the normal Hg levels in soil were 0.05–0.08 μg/g [36]. The data from Myanmar, the Philippines, and Thailand showed higher Hg levels than United States Environmental Protection Agency (US EPA) generic soil guidelines value (0–0.2 μg/g) [45].The study found that amalgamation process in ASGM areas contributed considerably to the Hg concentrations found. Therefore, people residing near the ASGM area could be impacted by Hg exposure during the amalgamation process.

### 3.4. Hg Concentrations in Plants

Plants are widely used as biomonitors for monitoring environmental Hg [46]. Rasmussen et al. found that among vegetative structures, the leaves contained the highest Hg concentrations [47]. In plants that absorb Hg primarily from the soil, Hg contents were found to be higher in the roots. Conversely, for the plants that adsorb Hg primarily from the air, Hg contents were found to be higher in the shoots and leaf tissues [48]. Some studies have reported that crops such as vegetables are the sources of Hg exposure for people living in Hg-mining areas [49]. The Hg contents found in plant samples from Indonesia, Myanmar, the Philippines, and Thailand are presented in Figure 8.

Some recent studies from Indonesia have reported high Hg concentrations of 1.4 μg/g dry weight (d.w.) in leaves from plants that grew near ASGM locations [50]. Similarly, contaminated forage plants (an edible animal feedstock) were found at a gold-mining site in Southeast Sulawesi Province, Indonesia [51]. Fresh forage plant samples from the Rarowatu and North Rarowatu Districts of Bombana were studied [51]. The sampling locations were divided into reference, mining commercial, and ASGM. The highest Hg content of 9.9 ± 14 μg/g d.w. was found in the ASGM area. The values in the commercial mining and reference areas were 3.20 ± 3.50 and 2.70 ± 2.80 μg/g d.w., respectively. According to the critical limits for Hg related to ecotoxicological effects on plants, the Hg levels in forage plants can be divided into three categories: high (>3 μg/g), low–moderate (0.1–3.0 μg/g), and low (0.1 μg/g) [51].

A study from the Mandalay region, Myanmar, included a preliminary survey that assessed the Hg air pollution in advance of future studies by examining tree bark, leaves, and blades of grass from *Typha latifolia* L. (leaf) species, *Azadirachta indica* (bark), *Terminalia catappa* L. (bark), *Manifera indica* L. (leaf), and *Naringi crenulata* (Thanaka (leaf)) [18]. Hg from the atmosphere and the soil can be deposited in plants [52]. The highest Hg concentration found was 4.17 μg/g d.w. in a Thanaka leaf near a gold shop, whereas the lowest concentration found was 0.02 μg/g d.w. in *Typha latifolia* L. leaf sample obtained some distance away from ASGM areas. The highest Hg concentrations found were in samples obtained near the gold refinery area.

Plants can serve as an indicator to regulate the uptake and transport of pollutants to the air because their internal pollutant concentrations are generally identical to the pollutant concentrations detected in the parent soil [53]. A study from the Philippines considered Hg concentrations in plant species including *Dadvalia* sp., *Alugbatging puti*, *Citrus* sp., cacao, and *Dilang aso* [27]. The study analyzed plant samples from an ASGM area and a few meters away from it. The Hg concentration in plants ranged between 0.04 and 34 μg/g d.w. [27]. The highest Hg concentrations was found in *Dadvalia*, which grew near a rod-mill station in the ASGM area [27].

In a study from Thailand, neem leaves and flowers from aquatic and terrestrial environments in the ASGM workplace in Phichit Province were investigated [31]. Neem flowers from aquatic sites showed Hg levels of 0.62–2.151 μg/g d.w. [31]. Similarly, neem leaves from terrestrial sites exhibited Hg levels in the range of 0.967 and 1.30 μg/g d.w.. The neem flowers were purposely collected from the aquatic tract, and the results showed that the highest Hg concentrations was found near the Hg emission source. Further, the study suggested that Hg concentration in neem flowers growing along the aquatic sampling site was related to the concentration of Hg in sediment at the same location [31]. The Hg levels were higher than the maximum permissible limit of Hg content (0.5 μg/g w.w.) for biota tissue [30]. Moreover, a study highlighted that the concentration of Hg can be deposited in legume species, such as *Indigofera enneaphylla* and *Desmodium triflorum* [31]. Therefore, the study suggested avoiding eating the plants near potentially Hg-contaminated areas.

Plants obtained in Indonesia, Myanmar, the Philippines, and Thailand have been found to contain Hg concentrations that were above the FAO/WHO guideline values (0.5 μg/g w.w.) [54,55].

### 3.5. Hg Concentration in Fish

Fish consumption is one of the most important factors contributing to Hg uptake in humans. Hg concentrations in fish tissues can be affected by the age, length, and weight of the fish [56]. Freshwater biota accumulate Hg from both natural and anthropogenic sources. Most fish have natural Hg levels of 0.02–0.3 μg/g wet weight (w.w.); however, small and short-lived herbivorous fish species have been found with a Hg level of 0.01 μg/g w.w. [57]. According to the recommendation of the FAO/WHO, the Hg content in a fish should not be more than 0.5 μg/g w.w. [58]. Hg concentration in fish from Indonesia and the Philippines based on w.w. are presented in Figure 9.

A study in Cambodia involved the collection of 82 fish species from local fishermen/fisherwomen in Kampi pool near Kraite, which is located near the O Tron gold-mining area [60]. The Hg concentration in the collected fish samples (*n* = 160) ranged between 0.008 and 0.64 μg/g. Additionally, the study grouped the size of fish as “small size” and “big size.” The big-sized fish showed an average Hg content of 0.128 μg/g, which was considerably higher than the average Hg of 0.086 μg/g in the small-sized fish [60]. However, the considered study had not determined the wet weight or dry weight for fish sample analysis. According to Baran et al. [61], Cambodians consume an average of 1.26 kg of fish each week; thus, Cambodians who consume more fish than the normal quantity intake higher Hg levels and are at a higher health risk [60]. 

Recent studies from Indonesia conducted fish sampling during 2007 to 2011 in Ratatotok Subdistrict, North Sulawesi, Indonesia, which was near the Mesel gold-mining area [59]. Local people from the studied area, which has an active fishing economy, have faced health issues during the period of active mining. The study involved the collection of fish samples from fishermen/fisherwomen and local market [59]. The fish samples from a Buyat Pantai fisherman showed Hg levels of 0.00–1.13 μg/g w.w. The samples from Buyat, Ratatotok, and Manado fish markets showed levels of 0.00–1.03, 0.00–0.53, and 0.00–0.17 μg/g w.w., respectively [59]. Thus, except for the Manado fish market, the fish samples from other sources exceeded the WHO standard guideline [58]. Nonetheless, the reported mean Hg concentrations in fish were within the standard limit for consumption.

Surveys were conducted to determine Hg concentration in marine samples in Davao del Norte, south of Manila, the Philippines, and near a gold-processing area. At the local market of Apokon, Tagum, seventeen specimens of fish and one seaweed sample were examined to determine the Hg and MeHg concentrations, which ranged between 0.001 and 0.44 μg/g w.w. and 0.007 and 0.38 μg/g w.w., respectively [37].

The considered studies revealed that the maximum concentration of Hg found in fish from Cambodia, Indonesia, and the Philippines were 1.13 μg/g, 0.44 μg/g w.w., and 0.64 μg/g w.w., respectively. Compared with the studies from Latin America, such as Brazil (1.04–2.84 μg/g w.w.), Colombia (1.60–4.50 μg/g w.w.), Bolivia (1.08–2.86 μg/g w.w.), and Ecuador (1.39–1.6 μg/g w.w.) [62], the reported Hg concentrations in Southeast Asia were relatively lower. 

### 3.6. Hg Concentrations in Human Hair

Hair is a common biomarker for characterizing MeHg exposures [63]. Low Hg concentrations have been considered risks for neurosis (50 μg/g) and health issues (11 μg/g) in unborn fetuses [64]. Moreover, a low Hg level in hair has also been associated with a low susceptibility of hair for Hg vapor. Hg concentrations measured in human hair from Cambodia, Indonesia, Myanmar, the Philippines, and Thailand are summarized in Table 4.

A study in Cambodia study involved the collection of human hair samples around the Mekong River, which is one of the world’s major rivers [60]. Hair samples were taken from people including mine workers living in the area of the O Tron gold mines and upstream and downstream along the Mekong River. These results revealed that the mean Hg concentration (5.21 μg/g) in the hair samples from men (*n* = 32) was higher than that of women (3.08 μg/g) (*n* = 46). When the female hair samples were sorted by sample area, the women from Ratanakiri province, near mine-affected areas, had a substantially higher Hg concentration of (3.47 μg/g) (*n* = 23) than a control group (2.7 μg/g) (*n* = 23) [60]. Hg levels in the Cambodian hair sample exceeded those observed near gold mines in the Philippines, where association of impaired human health with Hg concentration was observed [37].

A study involved the collection of human hair samples from an active ASGM area operating for more than 20 years in Lebaksitu, Indonesia [65]. The Hg hotspot village (Lebak-1) and downstream village (Lebak-2) were considered as high and low-risk areas, respectively. Human hair samples from both villages showed a mean Hg content of 3.2 μg/g, with a range of 0.847 to 9.015 μg/g [65]. The samples from Lebak-1 residents showed a considerably higher mean MeHg value (2.12 μg/g) than other residents, indicating that Lebak-1 residents were more exposed to MeHg than Lebak-2 residents. After comparison with other research on Hg-affected areas in Colombia [66], MeHg accumulation in hair from Indonesia was primarily caused due to consumption of food, such as fish and rice [65].

In the Mandalay region of Myanmar, human hair surveys were conducted of miners and nonminers living around the ASGM areas. The maximum Hg concentration in the hair samples of miners and nonminers were 5.7 and 2.9 μg/g, respectively [21]. The fact indicates that Hg concentration in human hair from the considered study was not at a level that would adversely affect human health because the approximate lowest levels of Hg that can cause neurosis and health problems in unborn fetuses are 50 and 11 μg/g, respectively [64,67].

Human hair samples from 70 inhabitants of Acupan, Benguet, in the northern regions of the Philippines, were obtained [68]. In the large studied ASGM community, the age of the participants was in the range of 8–66 years. The results showed that the average Hg content in the inhabitants was 3.47 μg/g. Hg concentrations in nine interviewees were higher than the human biomonitor limit of 5 μg/g [68]. Additionally, the highest Hg concentration was 26.6 μg/g, which was found in a 46-year-old male participant who was involved actively in amalgamation burning and lived only 5 m away from the ASGM location.

Hair samples were obtained from miners, schoolchildren, and a control group from the Phanom Pha gold-mining area located in Nong Pra subdistrict, Wang Sai Poon District, Phichit Province, Thailand [69]. The study considered the miners involved in the amalgamation process and working the ore-preparation area as groups I and II, respectively. The schoolchildren belonged to the group involved in gold-mining activities. The hair samples from miners showed an average Hg concentration of 1.17 (μg/g), which was within the reference group’s Hg concentration range [69]. The average Hg content in hair samples from schoolchildren in group I and II were 0.95 μg/g and 0.90 μg/g, respectively. Both schoolchildren groups showed Hg concentrations that were within the range of the control [69]. The fact suggests that lower Hg concentrations are expected in the hair because exposure to Hg is primarily due to inorganic Hg (i.e., Hg vapor) [69].

### 3.7. Health Risk Assessment of ASGM Communities

A study in Central Sulawesi, Indonesia, examined the health risks Hg exposure caused by the Poboya ASGM sites at the residential areas of Palu city [26]. The study focused on miners and other residents to estimate their health risk exposure. In each of the five locations studied, the frequencies of each hazard quotient (HQ) ratio (HQ ratio ≥ 1) from gaseous Hg (0) inhalation risk were determined [26]. Based on daytime Hg (0) concentrations, only 1.5% in the gold-processing area showed HQ ratios of <1, suggesting no risk. However, 93% of the sample population was found to be at risk. There are high chances of inhaling Hg released via ASGM activities in studied area. The human health risk from Hg exposure is particularly high in the Poboya gold-processing area and the areas close to Palu city. Moreover, 93% of the sample population in the Poboya area exceeded the no-risk values with HQ ratio > 1. These findings suggest that people who work in the gold-processing industry and nonminers in Palu city are at risk of adverse health impacts due to inhalation of Hg vapor. 

A preliminary health survey conducted in ASGM area of Thabeikkyhin Township, Mandalay region, Myanmar, involved the health inspection of men (*n* = 18) and women (*n* = 11) [19] to determine the health of the neurological system and respiratory functions. Based on the neurological assessment, three female miners who participated in ASGM panning and amalgamation processes for more than 5 years were diagnosed with mild tremors and ataxia. The respiratory assessment by spirometry on miners showed 38.9% normal, 27.8% mild, 27.8% moderate, and 5.6 % severe conditions. Meanwhile, the nonminer group exhibited 27.3% normal, 27.3% mild, and 45.5% moderate influence conditions [19]. Furthermore, the study found that with an increased duration of mining activities, the FEC and FEV1 values declined, indicating a chronic damage of the respiratory function in the enrolled miners. Therefore, health inspections of the ASGM community in Myanmar should be conducted intensively.

The health impacts of Hg on miners and children in the vicinity of ASGM mining in Apokon, Tagum, Davao del Norte, the Philippines, were studied [37]. The neurological effects found were mainly located on the cranial nerves (17.1%), reflexes (5.1%), sensory (5.1%), cerebellar (3.89%), and motor nerves (1.2%). The neurological effects were characterized as follows: cranial nerve VIII abnormalities (6.87%), distally decreased vibratory sense (2.69%), palmomental reflex deficiency (2.4%), cranial nerve I (2.40%), visual acuity (2.10%), and Babinski (1.50%). Based on physical examination, abnormalities were found in all 163 children enrolled in the study with the following five predominant abnormalities: below-average height, gingival discoloration, below-average weight, adenopathy, and dermatologic irregularities (Figure 10).

A study was conducted in the Phanom Pha gold-mining area of Thailand [69]. Two groups of miners and schoolchildren were divided into groups I (involvement in mining activities) and II (no involvement in mining activities) to estimate individual health risks. According to the U.S. EPA, the reference dosage is 0.0003 mg/kg/day [70], and the HQ ratio represents the estimated exposure intake. Low exposure to Hg vapor in the group of miners was evidenced by the range of Hg in the air of 0.005–0.021 mg/m^3^. The HQ ratios of group II indicated no risk [69]. However, the HQ ratios ranged from 16 to 218 in group I, which were much higher than in the HQ values of group II. Regarding the group of schoolchildren, group I exhibited a low HQ value of 0.02–0.23, while group II showed an even lower HQ value of 0.01–0.02 [69]. The higher HQ values of group I could be attributed to the Hg exposure from gold mining in the vicinity of amalgamation open burning. This suggests that the miners who work in the amalgamation process are at the greatest risk of inhaling Hg vapor. Therefore, miners who work in the amalgamation process are at the greatest risk of Hg vapor inhalation. Mitigation strategies to lower Hg contamination in the workplace must be considered.

## 4. Discussion

This paper reviewed the Hg pollution from ASGM areas in Myanmar and other Southeast Asian countries. Environmental indicators (e.g., air, water, and soil) and biomonitors (e.g., plants, fish, and human hair) were used in the considered studies. The concentrations of Hg in the air found at various areas in Indonesia, Myanmar, and the Philippines were higher than the standard limit values indicated in WHO guidelines [28]. The high Hg concentrations in the air were mainly due to the burning of gold amalgamation in the studies areas. By contrast, the reported concentrations of Hg in the air around ASGM areas in Idrija, Slovenia (<10 ng/m^3^) [71], and Guizhou, China (17.8 ng/m^3^) [72], were low. Moreover, in Almadén, Spain, where cinnabar was melted to produce Hg, the Hg levels reported were in the range of 100–14,000 ng/m^3^ [73], which were lower than the Hg levels in the ASGM areas in Myanmar and the Philippines.

In ASGM areas, water is essential for drinking and the domestic purposes of local people. Additionally, water purification is critical in the ASGM area because mine wastewater can be discharged directly into water bodies. Thus, the reviewed studies considered Hg concentrations in river water and groundwater around the ASGM areas. Hg concentrations in the water samples from Indonesia, the Philippines, and Thailand exceeded the WHO standard (0.5 µg/L) [40]. Hg concentrations in water samples from Myanmar was relatively lower compared with those from Indonesia, the Philippines, and Thailand.

Atmospheric deposition is the primary source of Hg in remote environments. Additionally, soil is another primary receiver of atmospheric Hg deposition in terrestrial ecosystems. Moreover, Hg can be retained by soil over long periods because of its elemental impurities [17]. Hg contents found in samples from Myanmar and the Philippines exceeded the standard limits of 1 μg/g in the U.S. (California), 6.6 μg/g in Canada, and 0.83 μg/g in the European Union (the Netherlands) [74].

Plants use their radicle system to absorb organic and inorganic Hg forms, which are then delivered to the leaves [75]. Temmerman et al. [76] found that Hg absorption also occurs through plant roots depending on soil exposure levels to Hg. Another theory is that Hg from the atmosphere can accumulate in most plants [77]. In this considered studies, Hg content in plants sampled from the Philippines showed the highest values. This was followed by Indonesia, Myanmar, and Thailand. Based on the findings of this review, the Hg levels found in the studied areas were higher than those reported in the Lanmuchang Hg-mining area, Guizhou Province, China (0.175 μg/g w.w.) [78]. Meanwhile, the reported Hg concentrations in vegetable samples collected at the Idrija Hg-mining area in Solvenia were <0.215 μg/g w.w. [71]. Compared with a study in the Alacran mine, Colombia [79], where the maximum value of Hg found in a leaf was 2.78 μg/g d.w., the values reported in the reviewed studies were higher. In addition, the Hg levels found in plant samples from the Almadén mining district, Spain, showed extremely high values in leaves, in the range of 0.16–1278 μg/g [80].

Fish in polluted water bodies are potentially contaminated by Hg. The reviewed studies investigated fish species obtained from local markets and fishermen within ASGM areas. Although the levels of Hg in some fish samples were below the WHO standard limits (0.5 µg/g w.w.) [58], the Hg levels in fish from fishermen sources in Indonesia and Cambodia were very high. Meanwhile, fish samples from the Philippines exceeded the U.S. EPA standard in fish tissue of 0.3 μg/g [62]. Generally, more than 75% of the Hg accumulated in the muscle tissues of freshwater fish is in the organic form of MeHg [81]. Moreover, seasonal variation such as precipitation should be considered, as there is a wide variety of aquatic habitats in the studied regions, which are affected by seasonal variations. For example, floods can temporarily modify the biogeochemical components (e.g., oxygen content, pH, and prey availability) of a system. Thus, the fish-sampling condition with respect to season (e.g., during dry or wet season) is important [82,83]. We therefore suggest that people living near ASGM areas should practice caution when consuming fish.

Mercury concentrations in human hair have been associated with both the endogenous Hg contamination through consumption of food that were contaminated by Hg species and the Hg concentrations in the air because elemental Hg can adhere to human hair [71,84]. The Hg concentrations in human hair samples from the studied ASGM areas in Cambodia, Myanmar, and Thailand were lower than those of residents of the Wuchuan Hg-mining area, China, (mean value and range of 34 and 7.6–93.1 μg/g, respectively) [78]. The lower Hg level may be attributed to lower susceptibility of human hair for Hg vapor [21]. In addition, Hg-mining areas in Guizhou Province and valley in the southern part of Shaanxi Province in central China showed Hg levels with mean values of 4.3 μg/g (1.6–12.6 μg/g) [85]. However, the Hg content found in the hair samples from the ASGM area in Lebaksitu, Indonesia, and the Acupan region, the Philippines, showed concentrations of 0.84–9.015 μg/g, and 0–26.6 μg/g, respectively [65,68]. Those value were above the allowable limit as per the WHO guidelines [67]. Meanwhile, the Hg concentrations found in Cambodia, Myanmar, and Thailand were within the recommended limit. In addition, a study reported links between high fish consumption and burning gold-amalgam exposure to high levels of Hg in human hair [86].

In an ASGM process, the final stage is the most critical in Hg inhalation because miners are exposed to Hg vapors during amalgamation burning. The average Hg concentration in the air of Palu city, Indonesia, was 12,782 ng/m^3^ [26]. The study indicated that 93% of the population was above the no-risk HQ ratio Therefore, both miners and nearby residents were at the risk of adverse health effects resulting from inhalation Hg vapor [26].

A study in Myanmar conducted a health inspection around an ASGM area in the Mandalay region. Based on the Human Biomonitoring Commission Standard, seven miners were in the range of warning status. Furthermore, the study highlighted that 16% of miners showed signs of Hg poisoning, such as nervous system damage, whereas nonminers did not demonstrate aberrant symptoms [19].

A study in the Philippines conducted physical examinations on 163 children [37]. All children showed the following common abnormalities: lower-than-average height, gingival discoloration, lower-than-average weight, adenopathy, and dermatologic abnormalities. According to the WHO, an adult who had an intake of 200 mg/day of Hg (e.g., from fish) has a 0.3% and 8% chance of experiencing paresthesia symptoms, respectively [87].

In the Phanom Pha gold-mining area, Thailand, the Hg exposure of miners and schoolchildren after ASGM activities was investigated. The HQ to a reference dosage (0.0003 mg/kg/day) was below the level at which unfavorable health effects on miners should be predicted [69]. The high-exposure miner group and schoolchildren showed the HQ ratios of 16–218 and 0.02–0.23, respectively. Inhalation of gold-amalgamation vapor can accumulate in the brain and kidneys [69]. Indeed, a study reported that miners in Brazil who used the open-burning method without using a Hg retort showed Hg levels that were higher than the normal Hg concentration in urine with using a Hg retort [69].

## 5. Conclusions

This review assessed and identified the Hg pollution in the ASGM areas of Myanmar and some other relevant Southeast Asian countries. Research should continue to focus on the current situation of ASGM activities in the studied countries and other parts of the world that allow ASGM activities because Hg is released by ASGM areas, which is a persistent and toxic global pollutant that can be transported through the atmosphere and deposited in terrestrial and aquatic ecosystems. The study aims to contribute to further research activities, such as health inspection and Hg management from ASGM areas because Hg is still used in ASGM activities. For example, Myanmar has not recognized the national Hg inventory, and its research activities are still limited. Additionally, Myanmar is still not a part of the Minamata Convention. Because gold prices remain high, despite countries such as Indonesia ratifying the Minamata Convention, Hg demand is still high in ASGM activities.

According to these reviewed studies, it is evident that Hg continues to cause contamination in the vicinity of ASGM areas, including nearby residential areas. Human epidemiological assessments on Hg-related diseases should be undertaken on a regular basis in the ASGM areas. Hg-contaminated areas should be controlled using reasonable regulations, policies, and frameworks. In addition, innovative Hg-free processing technologies and alternative economies should be introduced to support ASGM communities to reduce Hg emissions. Consequently, awareness of Hg problems can effectively reduce Hg pollution in ASGM communities.

## Figures and Tables

**Figure 1 ijerph-19-06290-f001:**
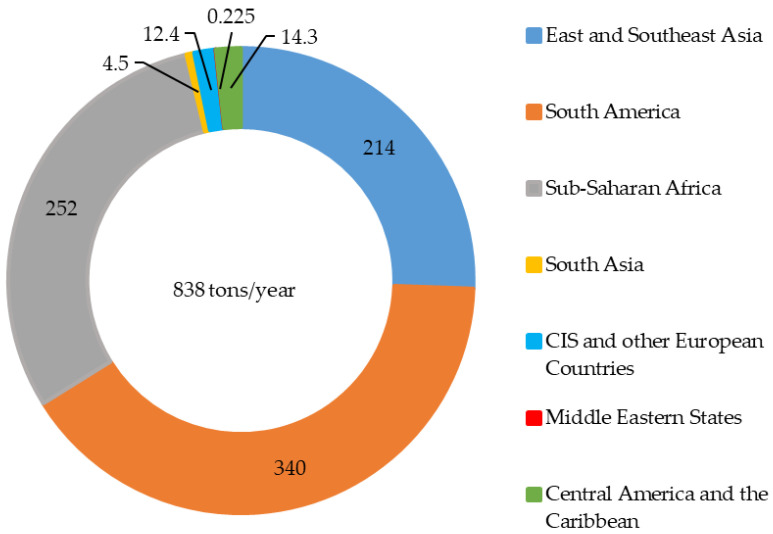
Regional results of global Hg emissions into air from the ASGM sources. Each value represents tons/year [4].

**Figure 2 ijerph-19-06290-f002:**
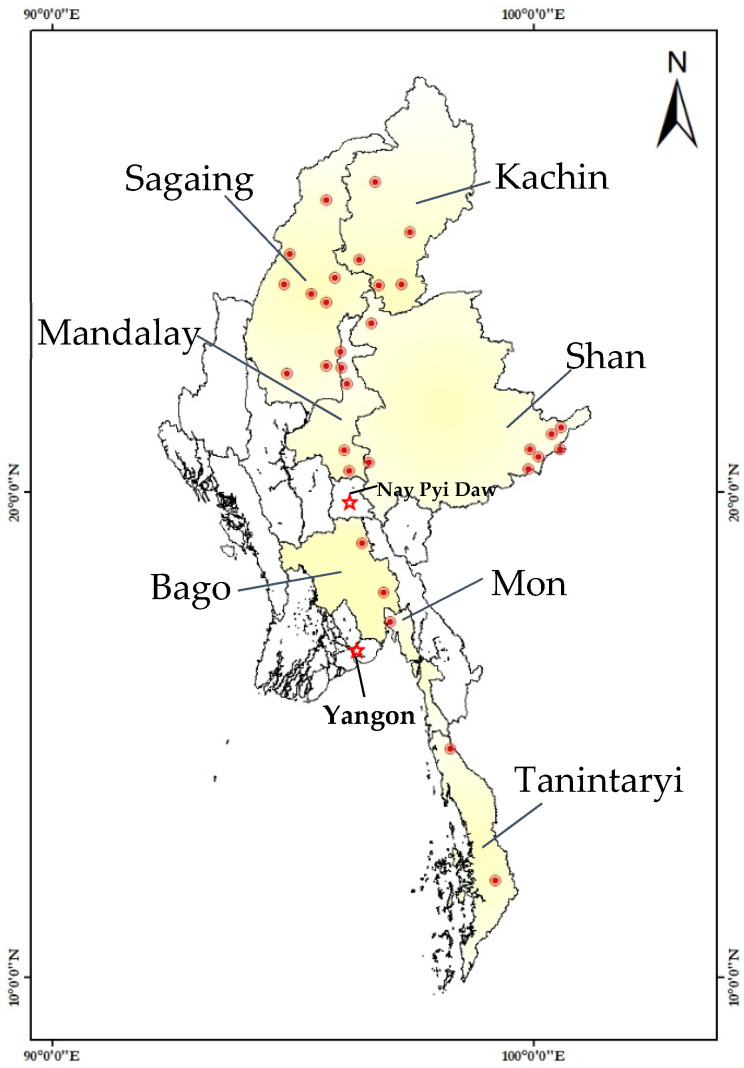
Distribution of artisanal and small-scale gold mining (ASGM) activities in the states and regions of Myanmar by township.

**Figure 3 ijerph-19-06290-f003:**
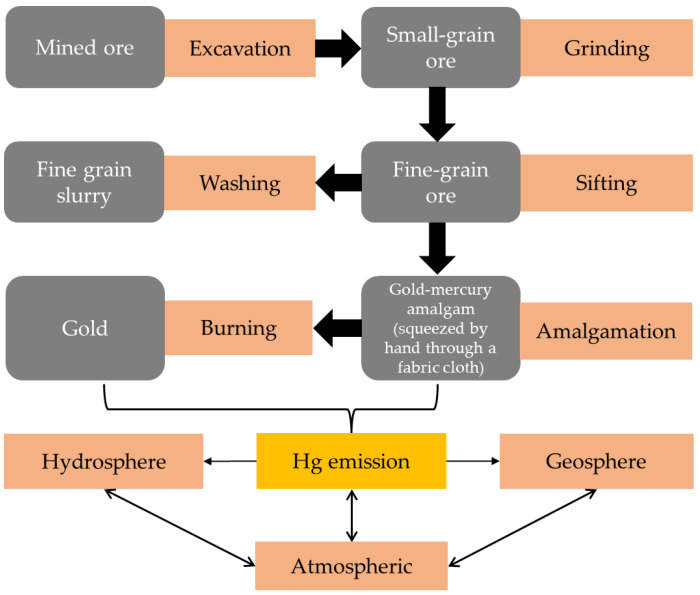
Flow chart showing general ASGM processes.

**Figure 4 ijerph-19-06290-f004:**
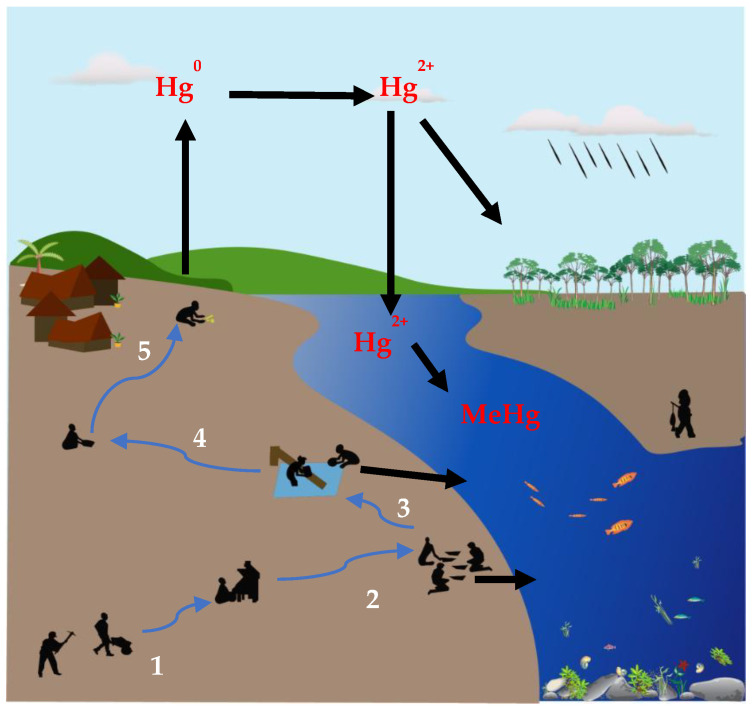
General ASGM processes involving (1) excavation, (2) grinding and sifting, (3) washing, (4) panning, and (5) gold–Hg-amalgam burning.

**Figure 5 ijerph-19-06290-f005:**
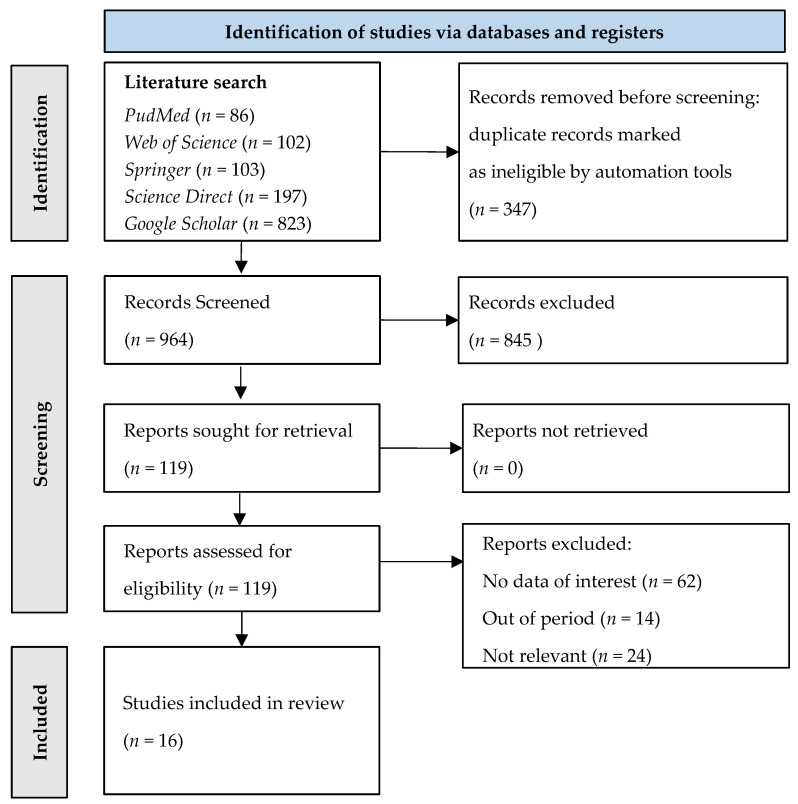
PRISMA flow diagram showing the search and selection process.

**Figure 6 ijerph-19-06290-f006:**
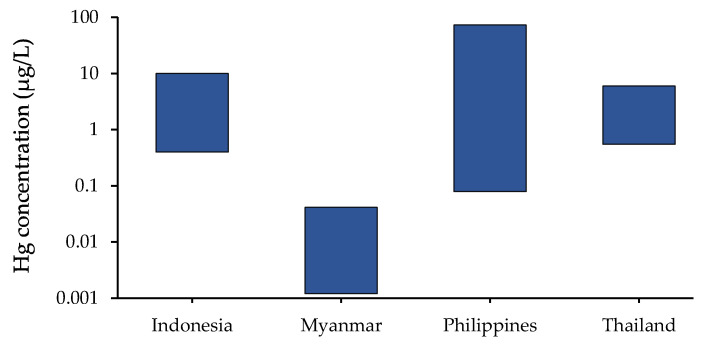
Hg concentrations in waterbodies from Indonesia [33], Myanmar [21], the Philippines [37], and Thailand [31]. Each piece of data represents a Hg concentration of range value, which was from a single study.

**Figure 7 ijerph-19-06290-f007:**
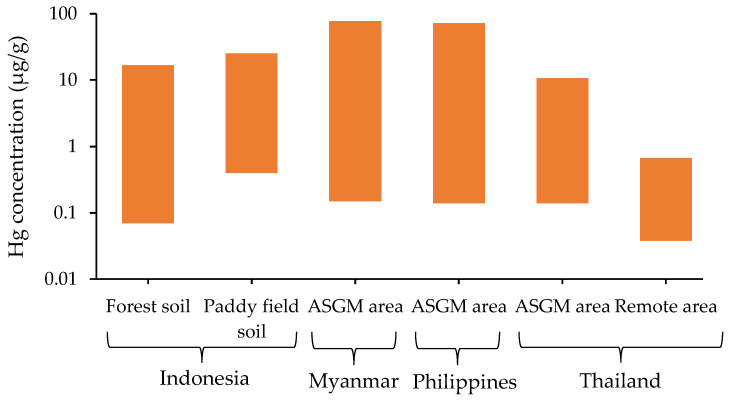
Hg concentrations in soil from Indonesia [33], Myanmar [17], the Philippines [27], and Thailand [31]. Each piece of data represents the Hg concentration of range value, which was from a single study.

**Figure 8 ijerph-19-06290-f008:**
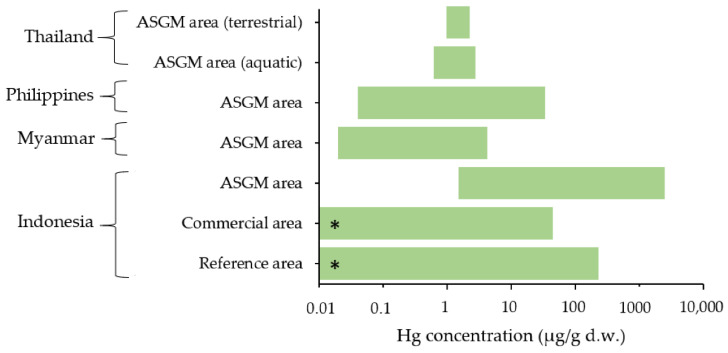
Hg concentrations in plants from Indonesia [51], Myanmar [18], the Philippines [27], and Thailand [31]. Each datum represents Hg concentration of range value, which was from a single study. *. minimum value of “0”.

**Figure 9 ijerph-19-06290-f009:**
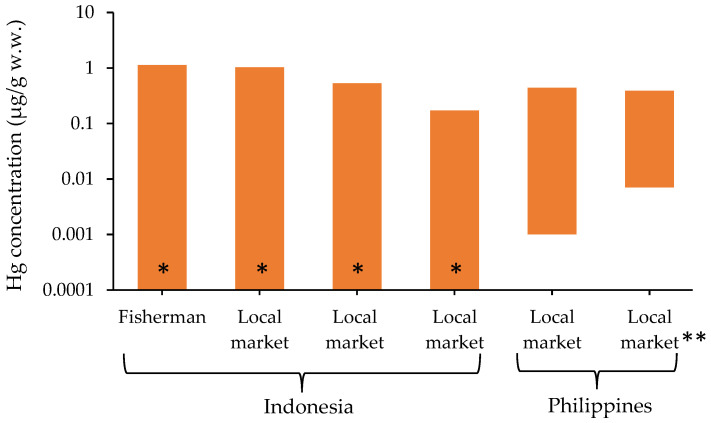
Hg concentrations in fish from Indonesia [59] and the Philippines [37]. Each datum represents Hg concentration of range value, which was from a single study. *; minimum value of “0”, **; MeHg concentration.

**Figure 10 ijerph-19-06290-f010:**
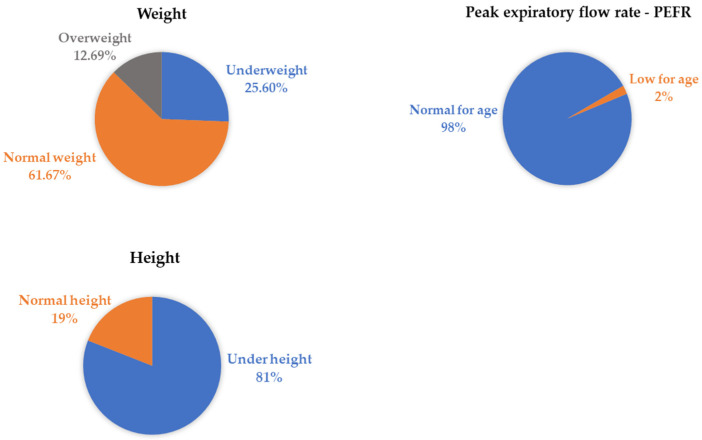
Predominant abnormalities found in schoolchildren of Apokon, Tagum, Davao del Norte, the Philippines [36].

**Table 1 ijerph-19-06290-t001:** Minamata Convention status in ASEAN countries [14].

Participants	Signature Date	Status	Date (Ratification/Accession/Approval)
Indonesia	10/10/2013	Ratification	09/22/2017
The Philippines	10/10/2013	Ratification	8/7/2020
Cambodia	10/10/2013	Ratification	8/4/2021
Vietnam	11/10/2013	Approval	06/26/2017
Malaysia	24/09/2014	Signature	
Laos		Accession	09/21/2017
Thailand		Accession	06/22/2017
Brunei	N.A.	N.A.	N.A.
Myanmar	N.A.	N.A.	N.A.

N.A.; not available.

**Table 2 ijerph-19-06290-t002:** Criteria for inclusion and exclusion of a study.

Inclusion	Exclusion
Studies related to ASGM communities.	Studies in other industries, such as coal-fired power plant.
Studies on Hg compounds, total Hg, inorganic Hg, and MeHg.	Unrelated compounds, such as ethylmercury.
Emphasis on Hg concentration in environmental media (e.g., air, water, and soil)	Measurement of Hg in other environmental media.
Emphasis on Hg concentrations in biomonitors of plants, fish, and human hair.	Measurement of Hg in other biological indicators.
Reports relating to human-health risk assessment in ASGM communities.	Reports relating to human-health risk assessment in other industries.

**Table 3 ijerph-19-06290-t003:** Summary of Hg concentrations in air from Indonesia, Myanmar, and the Philippines.

Location(s)	Sample Sources	*n*	Hg Concentration (ng/m^3^)	Reference
Palu city, Sulawesi, Indonesia	Gold-processing area	21	9172 ± 16,422 (mean ± SD)	[26]
	Northern area of city		514 ± 420 (mean ± SD)	
	Central area of city		141 ± 141 (mean ± SD)	
	Western area of city		22 ± 15 (mean ± SD)	
	Southern area of city		116 ± 135 (mean ± SD)	
Mandalay region, Myanmar	ASGM site	13	0–10,900	[21]
		19	0.66–74,000	
Province of Camarines Norte, The Philippines	ASGM site	4	7.8–314,000	[27]

N.A.; not available, S.D.; standard deviation.

**Table 4 ijerph-19-06290-t004:** Summary of Hg concentrations in human hair from Cambodia, Indonesia, Myanmar, the Philippines, and Thailand.

Location	Sample Source	Number of Samples	THg Concentration (μg/g)	MeHg Concentration (μg/g)	Reference
Mekong River, Cambodia (near O Tron gold mine)	Tonle Srepok	25	4.54 *	N.A.	[60]
	Tonle Kong	17	4.22 *		
	Mekong N. Stung Treng	16	3.36 *		
	Mekong Kratie	20	3.47 *		
	All males	32	5.21 *		
	All females	46	3.08 *		
	All adults	59	4.01 *		
	All children (aged < 13 y)	19	3.38 *		
	Women Ratanakirri (mine impacted)	23	3.47 *		
	Women Mekong	23	2.7 *		
Lebaksitu, Lebak regency Java Island, Indonesia	ASGM area	41	0.847–9.015	0.37–4.33	[65]
Mandalay Region, Myanmar	ASGM area (miners and nonminers)	50	0.4–5.7	N.A.	[21]
Acupan region, Benguet, the Philippines	ASGM area	70	0–26.6	N.A.	[68]
Nong Pra subdistrict, Wang Sai Poon district, Phichit Province, Thailand	Gold miners	79	1.17 ± 0.05 (mean ± SD)	N.A.	[69]
	School children	59	0.93 ± 0.01 (mean ± SD)		

N.A.; not analyzed, S.D; standard deviation, *; mean concentration.

## Data Availability

Not applicable.

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
