# Peer review of "Mercury Pollution from Artisanal and Small-Scale Gold Mining in Myanmar and Other Southeast Asian Countries"

_ijerph, 2022, doi:10.3390/ijerph19106290_

Round 1

Reviewer 1 Report

This is an interesting study to review and summarize Hg Pollution from Artisanal and Small-scale Gold Mining in Myanmar and other Southeast Asian Countries. The objectives of this study are clearly outlined, but there are some points that need be addressed before this manuscript could be considered for publication.

Abstract

You should describe how this review was conducted. You should follow a typical abstract format which includes a background, objective, method, results (main findings) and conclusion within limit word counts.

Introduction

Line 186-188: It should be revised to “The Hg-based amalgamation process used in ASGM sectors typically discharges wastewater into water bodies. Subsequently, the elevated Hg in water bodies can be uptaken by aquatic organisms”    

Conclusions

Rewrite your conclusions. It should provide a clear scientific justification for your work in this section and suggest future studies and point out those that are underway.

Table

Table 1: It is worthwhile to add another column to the right end for the reference of each country.

Table 2: It is better understood to prepare the location by countries in alphabetical order in the first column and references should be kept at the last column in the table.

Table 3: Same comments to Table 2

Figures

Figure 2: It is worthwhile to show the old and new capitals in Myanmar in the map.

Figure 6-9: The Y axis presents THg or MeHg? Clarify it.

Author Response

Dear Reviewer, 

Thank you very much for your valuable comments and suggestion. 

Please kindly see the attachment.

Reviewer 2 Report

Comments and suggestions see the attachment.

Author Response

Dear Reviewer,

Thank you very much for your valuable comments and suggestions.

Please kindly see the attachment.

Reviewer 3 Report

Comment to the authors:

This article aims to summarize mercury pollution from ASGM in Asian countries, analyze the mercury levels in environmental media such as air, water, and soil, as well as wildlife and human hair, and assess the health risk of humans related to mercury exposure. It is an interesting article that revises a number of relevant studies on mercury pollution from ASGM and adverse health effects linked to mercury exposure.

As a review article, I would suggest adding partial conclusion on the end of each section of results. For example, summary of metallic mercury emission from ASGM should describe at the end of 3.1 section. Also, the authors should briefly summarize the key points of available evidence in the section of conclusion. It is unnecessary to emphasize the aims of this study.

Line 239, Please check the concentrations “0-06.16.7”

Line 516, Please clarify “Another Hg mining area in Guizhou province in a valley in the southern part of Shaanxi province in central China was 4.3”

Author Response

(The authors gave the same response as above.)

Reviewer 4 Report

Dear Authors,

The problem of mercury pollution of the environment is significant due to its high toxicity to humans. At the same time, in some countries the gold mining process is conducted in a way that promotes the release of large amounts of Hg into water, air and soil. For this reason, the review of research on this problem for the countries of Asia SE should be considered important and necessary. Below are some comments that I believe may improve the quality of the manuscript.

  1. Why is Myanmar mentioned in the title if it is only one of the analyzed countries? The amount of data from Myanmar is not even greater.
  2. As it stands, the abstract resembles an introduction. I would suggest adding the most important data resulting from the review - for example on the level of pollution.
  3. I think that also in the case of the review paper, the Introduction should contain information on clearly defined research objectives.
  4. The description of database searches is not entirely clear. Were single or combinations of keywords used in the search? What does the term "original secondary information of relevant research studies" mean? What part of the publication came from this source?
  5. Why were so many publications excluded from the analysis? It seems that this should be explained a bit more.
  6. I wonder if it would not be worth presenting in which journals these 18 analyzed articles were published. Were these journals with an international scope, with an impact factor, or rather national journals?
  7. Perhaps a map should be included in the paper showing the spatial distribution of the research discussed in the paper.
  8. Line 172 - It is not entirely clear to me how to understand the statement that atmospheric mercury emissions in Malaysia are estimated at 0-21 mg / year. This is a gigantic difference, does this claim anything?
  9. Subsection 3.7. (lines 414-439). Where is this data coming from? No publication is cited here.
  10. Discussion - In my opinion, the summary of the research review is missing - what research gaps were found? - which of the components of the environment is the most polluted? - what research directions should be developed in the future. Perhaps it would be worth comparing the level of pollution with countries in South America and Africa where there are similar problems of Hg pollution.

Author Response

(The authors gave the same response as above.)

Round 2

Reviewer 1 Report

This revision is much more improved, but please double check the Hg concentration in Figure 6 if the title of vertical axis is correct. Following your response to reviewer's comment, it should be THg. 

Author Response

Dear Reviewer,

We appreciate your insightful suggestions greatly. We revised the axis concentration as "THg" in figure 6 of the manuscript. Please kindly see it.

Yours sincerely,